# Bio-Inspired Control System for Fingers Actuated by Multiple SMA Actuators

**DOI:** 10.3390/biomimetics7020062

**Published:** 2022-05-13

**Authors:** George-Iulian Uleru, Mircea Hulea, Adrian Burlacu

**Affiliations:** 1Department of Computer Engineering, Gheorghe Asachi Technical University of Iasi, 700050 Iasi, Romania; george-iulian.uleru@academic.tuiasi.ro; 2Department of Automatic Control, Gheorghe Asachi Technical University of Iasi, 700050 Iasi, Romania; adrian.burlacu@academic.tuiasi.ro

**Keywords:** anthropomorphic finger, multiple SMA actuators, spiking neural networks, biomimetic motions

## Abstract

Spiking neural networks are able to control with high precision the rotation and force of single-joint robotic arms when shape memory alloy wires are used for actuation. Bio-inspired robotic arms such as anthropomorphic fingers include more junctions that are actuated simultaneously. Starting from the hypothesis that the motor cortex groups the control of multiple muscles into neural synergies, this work presents for the first time an SNN structure that is able to control a series of finger motions by activation of groups of neurons that drive the corresponding actuators in sequence. The initial motion starts when a command signal is received, while the subsequent ones are initiated based on the sensors’ output. In order to increase the biological plausibility of the control system, the finger is flexed and extended by four SMA wires connected to the phalanges as the main tendons. The results show that the artificial finger that is controlled by the SNN is able to smoothly perform several motions of the human index finger while the command signal is active. To evaluate the advantages of using SNN, we compared the finger behaviours when the SMA actuators are driven by SNN, and by a microcontroller, respectively. In addition, we designed an electronic circuit that models the sensor’s output in concordance with the SNN output.

## 1. Introduction

The third generation of neural networks, which are implemented with spiking neurons, rigorously model the behaviour of the neural tissue. By introducing timing in information processing and adaptability these spiking neural networks (SNNs) are sensitive to time-varying functions and random occurrence of events [1,2]. Being based on the simultaneous operation of a significant number of neurons, the SNNs are most suited for hardware implementation which provides energy efficiency and real time response that does not depend on the number of neurons [3,4]. These properties of analogue hardware constitute critical advantages to modelling the brain functions using SNNs [5].

The main characteristics of the spiking neurons relate to spike processing which includes spatial and temporal integration of the incoming stimulation, detection of activation threshold, synaptic delay, refractory period, and generation of excitatory or inhibitory pulses [6].

Being the most rigorous model of biological neural networks, there has been a growing interest in the use of SNNs in a wide range of bio-inspired applications. Recently, superior abilities of the brain such as symmetry perception [7], visual pattern recognition [8], and speech recognition [9,10] were modelled using neuromorphic hardware based on spiking neurons. In robotics, different types of neural systems were designed for the control of vehicles speed [11] or trajectory [12], as well as for modelling the motion abilities of the human body including the control of robotic arms [13,14,15] or anthropomorphic fingers [16,17].

Due to its impressive complexity and dexterity, the human hand is used as an iconic model for the development of different robotic hands that can adapt to a wide variety of gasping and manipulation scenarios. Modelling and actuating anthropomorphic fingers has been a field of study for a long time with lots of effort invested, having as the main characteristics the implementation level of the biological features and the actuation method [18,19]. Some of the previous work is focused on proposals with a very high degree of similarity compared with the biological model [20,21], while others proposed new methods to reduce the number of artificial tendons using mechanical workarounds such as pulleys [18] or differentials [22]. DC motors are the most common devices used for actuation [23,24] which are connected to the phalanges with wires [25] or rods [26]. Currently, there is an increasing trend to use flexible actuators such as pneumatic tubes [20,27] and actuators made of shape memory alloy (SMA) [28] to implement soft actuation methods [29]. In addition, most of the systems are coupled or under-actuated for using a lower number of control inputs than the degrees of freedom [29]. One such example is an advanced prosthetic hand actuated by DC motors that are controlled by EMG signals to perform semi-autonomous grasping [30].

### 1.1. Biological Background

A fundamental question of scientific and clinical importance is how the brain and spinal cord control the muscles. A widely used experimental technique that can provide an answer is to infer the neural control methods by analyzing samples of muscle activity and limb mechanics collected while animals and people are in motion. The human hand has a very large number of degrees of freedom and adaptability that are difficult to control. Studies of the physiology of cortical and spinal neurons as well as electromyographic (EMG) activity of muscles have led to a popular, but not yet proven, hypothesis that the motor cortex and spinal cord simplify the control of the numerous muscles by grouping them into few functional units called neural synergies [31,32]. In typical activities, the human hand uses two or more fingers with more joints to achieve desired actions including net flexion or extension force generation. The use of multiple fingers implies that individual joints involved in the activity work together for the successful completion of the tasks making the hand an excellent example of kinetic redundancy. This implies that the number of elements (finger joints) is larger than the number of constraints creating a use-case for neural synergies [33,34].

In addition, the high complexity motions of the human hand involve lateralized activation of the motor cortex which implies motor planning for the active hand and inter-hemispheric inhibition. Note that the loss of this lateralization determines involuntary symmetrical movements of one side of the body that mirror voluntary movements of the other side [35].

Note that lateralization represents a basic element in the brain organization from the small brains of insects to variously sized brains of vertebrates, including humans. This characteristic implies that the left and right sides process information differently and control different patterns of behaviour. Lateralized brains can carry out different functions in parallel on the left and right sides avoiding duplication and increasing cognitive capacity [36]. Recent research highlights the important role of lateralization played in the development of biorobotic artefacts of high biological plausibility [37].

### 1.2. Biomimetic Control Methods for Robotic Hands

Among the control techniques, SNNs are used for fingers that are actuated by motors in several grasping activities that depend on the object type. In this case, the simulated SNNs use several hundred neurons for the control of each finger of anthropomorphic hands [17]. In contrast, our previous work demonstrates that less than 50 neurons implemented in analogue hardware are able to control the rotation of a single joint robotic finger actuated by a shape memory alloy (SMA) actuator [16]. Based on this control principle that was demonstrated for the single actuator, in the current research we evaluate the SNN’s ability to control in sequence the multiple SMA wires that actuate all junctions of the index finger. In order to obtain biomimetic motions, the finger is flexed and extended by two pairs of SMA actuators that play the role of the main tendons of the index finger. The actuation sequence is obtained by driving the actuators involved in the subsequent motions when the output of the flex sensors reaches predefined thresholds that are set empirically. Taking into account that a simple method to drive the SMA actuators is by using continuous signals that typically are generated by microcontrollers (µC) [38,39], we comparatively evaluated the control methods based on SNN µC and on an SNN, respectively.

The contributions of our work are as follows:-the design of an SNN architecture that is able to control the sequence of actuators of an anthropomorphic finger;-experimental validation of SNN superiority in comparison with a microcontroller based control approach.

The remainder of the paper is organized as follows: in the next section, the system structure and the experimental setup are presented followed by the results and discussions in Section 3. Finally, the conclusions and future work are presented in the last section.

## 2. Materials and Methods

The proposed method based on an SNN for the control, in sequence, of the actuators that drive an anthropomorphic finger, was evaluated starting from simulations of the neural network activity. Following this preliminary phase, the physical implementation of the finger was controlled by hardware implementation of the neural network and by a microcontroller for comparison purposes.

### 2.1. System Structure

The bio-inspired system that controls the anthropomorphic finger includes the analogue SNN that is interfaced with the sensors and the SMA drivers.

#### 2.1.1. Mechanical Implementation of the Anthropomorphic Finger

According to the anatomy of the human hand, the index finger is driven by several tendons that are connected to the muscles. Thus, the finger is flexed by tendons FDP and FDS, which actuate the DIP and PIP junctions moving the distal and middle phalanges [40]. At the base of the finger base, the proximal phalanx is actuated by the tendon RI, determining the rotation of MP. Taking into account that for low forces which occur in unloaded finger’s motion, the FDS is not active [40], we implement only the actuators that model FDP and RI as presented in Figure 1a. This approximation is also in concordance with a more recent approach to implementing a robotic finger that used two tendons for flexing [18].

Similarly, for modelling the activity of EC and TE we implemented two artificial tendons for finger extension that actuate MP and, respectively DIP and PIP junctions, see Figure 1b. Thus, DIP and PIP are actuated simultaneously for both flexion and extension while MP is actuated independently, mimicking the behaviour of the index finger [27].

The rotation of the active junctions PIP and MP is converted into voltage by two resistive flexion sensors and the corresponding amplifiers. These sensors provide feedback about the finger motions which is critical for the sequential actuation of the SMA wires. Among the possible motions of the fingers [40] we considered for this work only the flexion and extension because their importance in object manipulation is significantly higher than the motions towards the sides such as abduction and adduction.

#### 2.1.2. Spiking Neural Network

The SNN is based on a neuron model that was previously presented in [16]. Here we focus only on the critical elements that are used to drive the SMA actuators according to the input signals. As shown in Figure 2, these neurons include one SOMA and at least one synapse (SYS) which can be excitatory or inhibitory. The synapses generate positive or negative spikes taking as reference the equilibrium potential $V_{EQU}$ of the SOMA. The spikes are integrated by the circuit INT which determines the activation frequency of the SOMA and, consequently, of the neuron. During each activation of the SOMA, all synapses connected to the hardwired axon generate a spike at their output $N_{OUT}$. The spike energy depends on its amplitude and duration which varies according to the synaptic weights stored by the synapses using capacitors [16]. In this work all synapses are excitatory and the weights are set to their maximum values to ensure the fastest response of the SMA actuators.

The input neurons of the SNN are activated by a constant voltage $V_{F}$ that determines the variation of the potential $V_{S}$ at the SOMA’s input. When $V_{F}$ is above $V_{EQU}$ the neuron activates with the frequency $f_{N}$ determined by $V_{F}$ and $R_{F}$. The postsynaptic neurons (posts) that receive pulses from other presynaptic neurons (pres) are activated when the potential integrated by the circuit INT is above $V_{EQU}$. Note that the input neurons include $R_{F}$ without INT, the integrator being used only with posts when the switch $SW_{F}$ disconnects $R_{F}$. The signal $V_{SPK}$ is used for monitoring the neuron activity [16] because the variation of $V_{SPK}$ during neuron activation mimics the postsynaptic membrane potential of the natural neurons.

#### 2.1.3. The Synaptic Configuration

The neural network uses a basic SNN (BSNN) for driving each actuator. The BSNN includes an excitatory neuron *E* that activates the motor neuron M through the integrator INT as in Figure 3a. The inputs of the BSNNs are activated by the control unit (CU) using push buttons (PB) or sensors. The CU includes adjustable resistors $R_{ADJ}$ or Zenner diodes to adapt the potential to the input of the neurons. Therefore, the CU generates voltage levels $V_{F}=V_{FDP},V_{RI},V_{TE},V_{EC}$ (see Figure 3b) which are converted into spiking frequency by the neurons *E* of the corresponding BSNN. Note that the spiking frequency of the neurons determines the contraction force $F_{S}$ of the SMA actuators implying that $F_{S}$ is directly determined by $V_{F}$. In addition, the motions which involve more SMA actuators are obtained when the control unit generates sequences of the potentials in the set $V_{F}$. The dependency between the junctions’ rotation is implemented in the CU by the connection of the basic SNN for initial and subsequent motions to the PB and to the sensor that detects the initial motion. In this setup, the proximal phalange is always actuated before the middle and distal phalanges to avoid exerting force on the proximal phalange when it is passive.

The schematics of the sensor amplifier AMP and the p-MOS circuit that drives the SMA actuators are given in our previous work [16].

### 2.2. Finger Motions

In order to evaluate the ability of the SNN to control the biomimetic motion of the robotic finger, we evaluated the junctions rotation during contraction of different combinations of SMA actuators type Flexinol LT of 150 µm width. The experiments are based on the biological evidence related to the possible motions of the human fingers focusing on the tendons involved in each motion [20]. Taking into account the physiology of the human finger [40], the rotation of PIP and DIP is initiated at the same time. For this work, the distal and middle phalanges are actuated by the same tendon. The bio-inspired motions denoted by #A, #B, #C, and #D and named according to [20] are presented in the Table 1 together with the involved tendons.

The drawings of the finger in the final positions are shown in Figure 4. During motion #A the actuators EC and TE are actively extending all phalanges of the finger while, in the opposite direction, the motion #B involves both flexor tendons RI and FDP for the full flexion of the finger. The extension of the proximal phalange by the actuator EC while the other phalanges flex due to the actuation of FDP corresponds to the motion #C. Another motion (#D) is performed by actuation of RI and TE when the proximal phalange extends while the middle and distal phalanges flex.

## 3. Results

The behaviour of the controlled finger is evaluated both in simulation and real experiments. A comparison is done between two control architectures first based on hardware SNN and the second based on a microcontroller.

### 3.1. Simulation of the Finger Motions Driven by SNN

A direct evaluation of the performance of the SNN in controlling the motion of the robotic finger can be obtained by modelling the influence of the SMA actuators on the joints rotation using the circuit from Figure 5. This circuit that is denoted by SMC receives input from the motor neurons and generates a function that mimics the output of the flex sensors. The capacitors accumulate the spikes delivered by the motor neurons that drive the flexors (RI and FDP) and extensors (EC and TE). The current sources are controlled by the voltage, which increases or decreases the potential in the capacitor $C_{p}$, simulating the sensor output when the corresponding SMA actuator contracts. The SMC parameters are adjusted empirically based on the sensor’s output when the finger is driven using continuous signals generated by the microcontroller. According to Figure 6a, the rotation of the robotic joint is accelerated in the first phase of the SMA contraction followed by a phase where the speed decreases until the rotation stops. The potential generated by SMC mimics the variation limits of the sensor’s output, as well as the exponential increase after the motion onset.

Using SMC, the rotation of each junction that is controlled by SNN was evaluated by computer simulations in LT Spice. The simulated output of the flex sensors that is determined by the motions #A, #B, #C, and #D is presented in the first column of Figure 7. These signals represent the ideal case of the finger motion that is compared in sequence with the real finger behaviour.

### 3.2. Motion Control Using the Hardware SNN

The performance of the SNN hardware implementation to control the motion of the anthropomorphic finger was analyzed via experimental results. The input potentials for the SNN that are generated by sensors or by pressing the PBs are presented in the Table 2. These potentials that actuate the corresponding SMA wires were determined empirically based on observations of the finger motion. As presented in the middle column of Figure 7, the variation of the sensor’s output matches qualitatively the simulated potentials in the first column.

The operation of the motor neurons $M_{EC}$ and $M_{FDP}$, included in the basic SNNs that drive the actuators EC and FDP, is exemplified in Figure 8. The magenta and green signals represent the $V_{SPK}$ potentials generated by the SOMA (see Figure 2) that vary when the neurons activate. The blue and red signals represent the AMP output $V_{AO}$ (see Figure 3a) showing the finger motion. The variation of AMP output highlights the delayed rotation of the junction MP after PIP. This delay is determined by the fact that the neuron $M_{FDP}$ activates after $M_{EC}$ when the sensor output (red signal) reaches the preset threshold $V_{EXT}$ (see Table 2).

### 3.3. Motion Control Using the Microcontroller

A typical method to control SMA actuators is based on continuous signals that are generated by microcontrollers (uC). Starting from this aspect we evaluate the behaviour of the finger when a uC drives in sequence the two SMA actuators involved in each motion. The SMA wire that drives the PIP joint is actuated when the output of the sensors for MP reaches predefined thresholds that are detected by uC using ADC. The results for this experimental phase are highlighted in the last column of Figure 7.

Note that the sensor’s output oscillates, implying that the finger speed varies during motions for both SNN and uC. Excepting the motion #B, these oscillations are similar for the two control methods implying that the speed variation during motions is determined mainly by the actuators and finger implementation. This aspect is sustained by the smooth variation of the simulated signals for the sensor’s output which are presented in the first column.

In addition, we used an electronic scale to measure the force of the finger during flexion when both SMA actuators are active and their supply current is limited to 400 mA. The obtained values are 178 g and 184 g, respectively, when the SNN and the microcontroller are used, implying that using continuous signals instead of spikes slightly increases the force of the finger.

## 4. Conclusions

Taking into account that the motor cortex uses neural synergies to control multiple motions of the human hand, we implemented a spiking neural network that activates groups of neurons to control sequences of motions of an anthropomorphic finger. The flexion and extension of this finger are determined by four SMA actuators that play the role of the main tendons of the index finger. The force of each SMA actuator is controlled by an SNN with a few excitatory neurons for which the firing rate is determined by the input voltage levels. For stimulation of the SMA actuators involved in the finger motions, the SNN inputs can be activated by PB for initial motions of the phalanges and by sensors for the subsequent motions. To evaluate the importance of using electronic neurons, the finger motions are evaluated comparatively when the actuators are driven using an electronic SNN or a microcontroller.

The results show that a simple SNN is able to smoothly drive an anthropomorphic finger in several biomimetic motions that can involve the activity of two actuators that contract in sequence. In addition, the finger motion is slightly smoother for some finger motions when the SNN is used. These results are encouraging to use of analogue implementation of spiking neurons and SMA actuators in motion control of robotic hands. For future work, we intend to make the SNN able to learn to actuate SMA wires in parallel or in sequences that are specific to more complex biomimetic motions. In the long term, the system can be significantly improved to use non-invasive EEG sensors that control the motion of an anthropomorphic hand in concordance with brain activity.

## Figures and Tables

**Figure 1 biomimetics-07-00062-f001:**
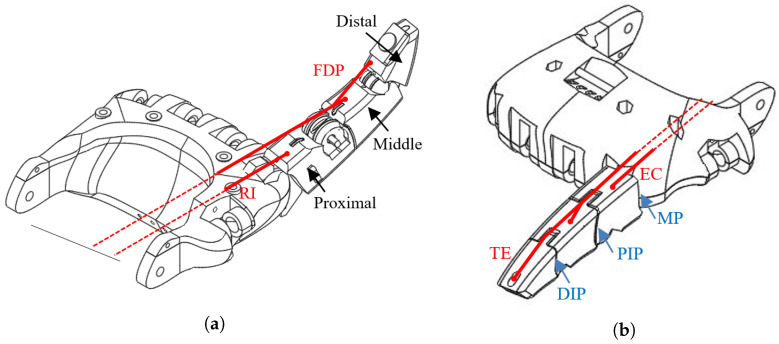
Mechanical structure of the artificial finger that is driven by the SMA wires: (**a**) FDP and RI for flexion and (**b**) TE and EC for extension.

**Figure 2 biomimetics-07-00062-f002:**
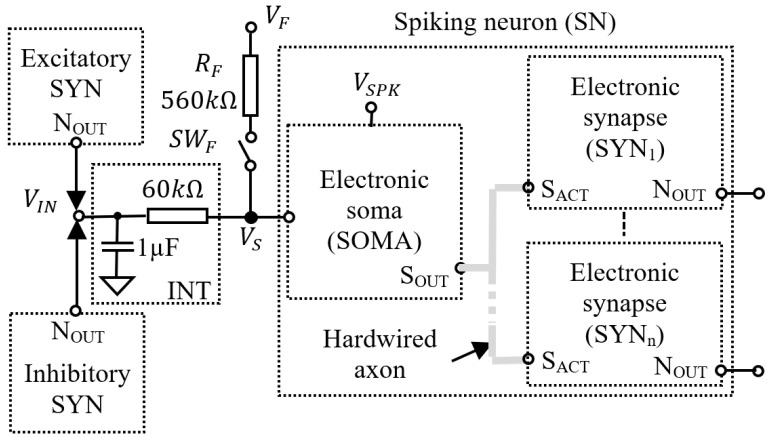
The structure of the electronic neuron that includes one SOMA and more synapses $SYN_{1},\cdots ,SYN_{n}$ with n≥1.

**Figure 3 biomimetics-07-00062-f003:**
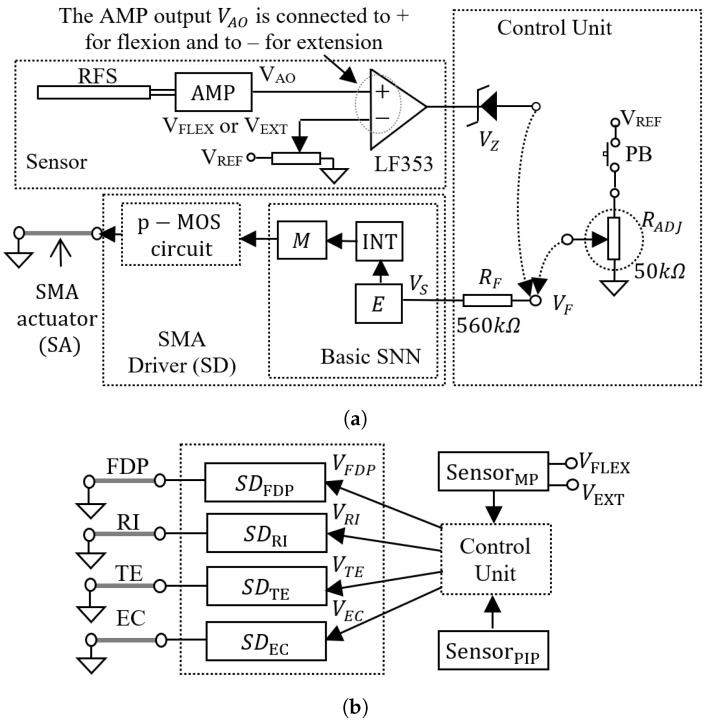
The structure of the SMA driver (SD) for a single SMA actuator control unit (CU) (**a**); the structure of the system that includes a SMA driver for each of the actuators FDP, RI, TE, and EC (**b**).

**Figure 4 biomimetics-07-00062-f004:**
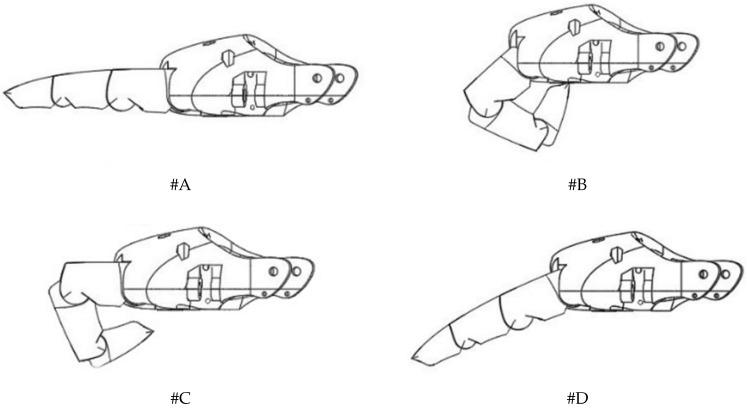
The final positions of the robotic finger for the considered motions #A–#D, according to the description in Table 1.

**Figure 5 biomimetics-07-00062-f005:**
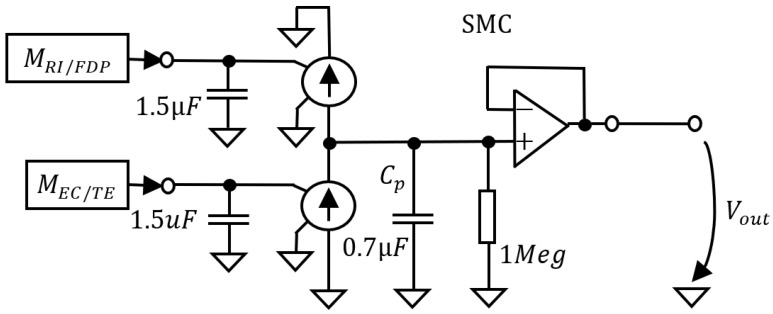
The circuit for modelling the joint rotation when the junction is actuated by SMA wire.

**Figure 6 biomimetics-07-00062-f006:**
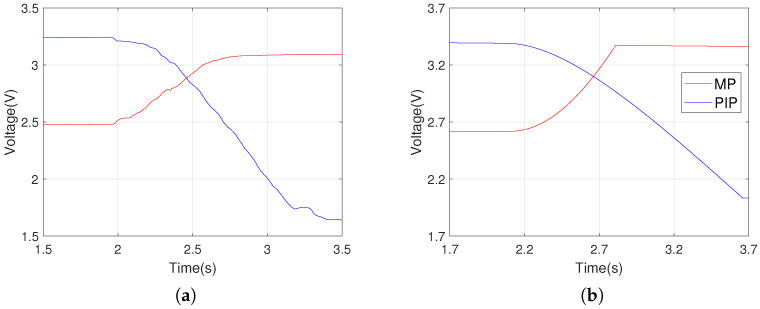
The similarity between (**a**) the output of the flex sensors when the finger is actuated by microcontroller and (**b**) the output of the circuit SMC that models the rotation of the robotic joint when actuated by the SMA.

**Figure 7 biomimetics-07-00062-f007:**
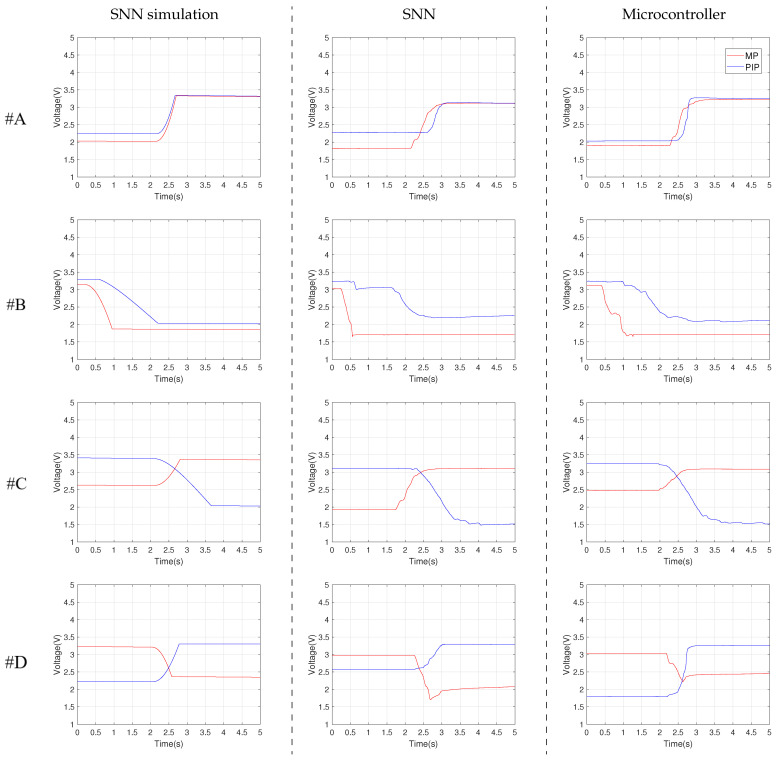
The output of the flex sensors for the considered motions #A–#D, according to the description in Table 1. The columns represent the experimental method used, while each line represents a particular finger movement.

**Figure 8 biomimetics-07-00062-f008:**
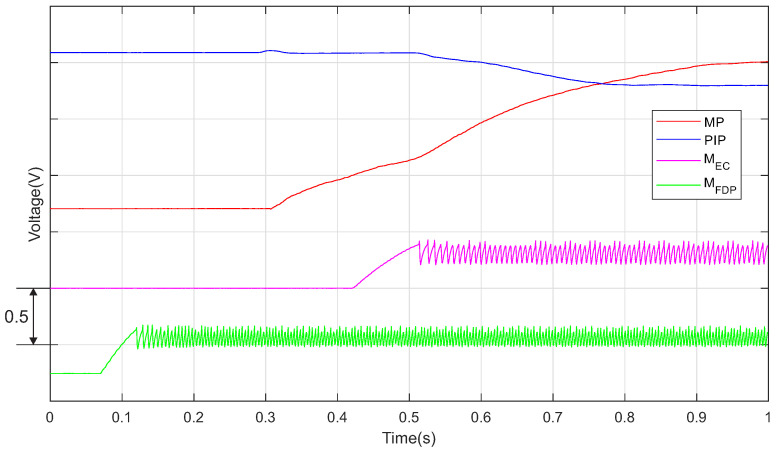
The activity of the motor neurons that drives the actuators EC and FDP together with the output of the sensors.

**Table 1 biomimetics-07-00062-t001:** The bio-inspired motions of the robotic finger.

Code	Motion Name [20]	Tendons
#A	Isometric MP extension	EC, TE
#B	Isometric interosseous	RI, FDP
#C	Isometric MP flexion	EC, FDP
#D	Isometric extensor	RI, TE

**Table 2 biomimetics-07-00062-t002:** Potential that we used in the experiments.

Parameter	$V_{\mathrm{RI}}$	$V_{\mathrm{EC}}$	$V_{\mathrm{FDP}}$	$V_{\mathrm{TE}}$	$V_{\mathrm{FLEX}}$	$V_{\mathrm{EXT}}$
Voltage (V)	2.5	2.5	2.8	2.8	2.8	2.0

## Data Availability

Not applicable.

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
