# Peer review of "Bio-Inspired Control System for Fingers Actuated by Multiple SMA Actuators"

_biomimetics, 2022, doi:10.3390/biomimetics7020062_

Round 1

Reviewer 1 Report

In this manuscript, the authors presented a SNN control for robotic finger driven by SMAs. Overall, the novelty is clear and the paper is well-written. The reviewer can recommend for publication with a minor comment. The Figure 6-8 can be improved. Currently, these figures has no axis ticks so the values of the data points are all unclear. Also, it seems that the colors of the curve in one figure is not consistant, for example, in Figure 6, the MP is red in (a) but is more like magenta in (b). This also happened in Figure 7, please check.

Author Response

Dear distinguished Reviewer,

We would like to deeply thank you for your very constructive comments. They have very helpful for revising our paper and allowed us to improve the general quality of the paper. We appreciate your efforts very much. We have taken into consideration and complied carefully with all the comments and suggestions.

Respectfully yours,

George Iulian Uleru

Mircea Hulea

Adrian Burlacu

Reviewer 2 Report

The manuscript is quite interesting but several issues need to be addressed. In particular authors have to further stress the biomimetic aspect of their study, as this is crucial for this Journal. Just reporting that SNN and/or anthropomorphic grippers have been used is not enough.

Furhtermore, authors may expand the part related to bioinspred control of biological-like grippers by intriducing the concept of laterality and lateralization  of brain and behaviour that is crucial in biological models to optimize tasks. Some relevant examples that would increase the scientific value of this work

Romano, D., Benelli, G., Kavallieratos, N. G., Athanassiou, C. G., Canale, A., & Stefanini, C. (2020). Beetle-robot hybrid interaction: Sex, lateralization and mating experience modulate behavioural responses to robotic cues in the larger grain borer Prostephanus truncatus (Horn). Biological Cybernetics114(4), 473-483.

Rogers, L. J. (2021). Brain lateralization and cognitive capacity. Animals11(7), 1996.

A deep English revision is needed.

Author Response

Dear distinguished Reviewer,

We would like to deeply thank you for your very constructive comments. They have very helpful for revising and improving our paper and allowed us to improve the general quality of the paper. We appreciate your efforts very much. We have taken into consideration and complied with all the comments and suggestions. We have studied the comments carefully and provided detailed explanations for our changes in this response. 

Respectfully yours,

George-Iulian Uleru,

Mircea Hulea

Adrian Burlacu

Round 2

Reviewer 2 Report

Authors addressed almost all my suggestion and the manuscript is now improved.

Maybe they can consider to also include these relevant researches to their study to improve its scientific value

Romano, D., Benelli, G., & Stefanini, C. (2019). Encoding lateralization of jump kinematics and eye use in a locust via bio-robotic artifacts. Journal of Experimental Biology222(2), jeb187427.

Youssef, I., Mutlu, M., Bayat, B., Crespi, A., Hauser, S., Conradt, J., ... & Ijspeert, A. (2020). A neuro-inspired computational model for a visually guided robotic lamprey using frame and event based cameras. IEEE Robotics and Automation Letters5(2), 2395-2402.

English still needs some revision.